# Effects of Low-Fat Distillers Dried Grains with Solubles Supplementation on Growth Performance, Rumen Fermentation, Blood Metabolites, and Carcass Characteristics of Kiko Crossbred Wether Goats

**DOI:** 10.3390/ani12233318

**Published:** 2022-11-28

**Authors:** Khim B. Ale, Jarvis Scott, Chukewueme Okere, Frank W. Abrahamsen, Reshma Gurung, Nar K. Gurung

**Affiliations:** Department of Agricultural and Environmental Sciences, College of Agriculture, Environment and Nutrition Sciences, Tuskegee University, Tuskegee, AL 36088, USA

**Keywords:** low fat, distillers dried grains with solubles, castrated goats, performance

## Abstract

**Simple Summary:**

Distillers dried grains with solubles (DDGS) is a byproduct of bioethanol industry, and it contains crude protein of 25% to 31% and ether extract of 9% to 13%. DDGS produced from bioethanol industry in United States (US) are predominately low fat (3–5%). Nutritional profile of low-fat DDGS (LF-DDGS) showed that it is suitable feedstuff for livestock including goats and may reduce the production cost. However, there are limited studies on the effect of low-fat DDGS (LF-DDGS) in goats. Thus, this study aimed to evaluate the effect of feeding different amounts of LF-DDGS on growth performance, growth efficiency, rumen fermentation, blood metabolites, and carcass characteristics of Kiko crossbred wether goats. The study was conducted for 84 days, feeding different amounts of LF-DDGS. The results suggest that up to 30% LF-DDGS can be included in diets of castrated male goats without affecting the growth performance and carcass characteristics.

**Abstract:**

Distillers dried grains with solubles (DDGS) produced in US are predominately low fat, as the economics favor separating as much oil as possible for sale as renewable diesel feedstock and also for use in swine and poultry feed. This study aimed to evaluate the effect of feeding different amounts of low-fat DDGS (LF-DDGS) on growth performance, growth efficiency, rumen fermentation, blood metabolites, and carcass characteristics of Kiko crossbred wether goats. Twenty-four goats, 5–6 months of age, were randomly assigned to one of the four experimental diets (n = 6/diet), 0%, 10%, 20%, and 30% LF-DDGS on an as-fed basis, and fed for 84 days. Data collected were analyzed using an orthogonal contrast test for equally spaced treatments. Average total gains, average daily gains, and gain-to-feed ratios were similar among the treatments (*p* > 0.05). Rumen acetate, propionate, and butyrate concentrations and acetate: propionate ratios were similar (*p* > 0.05) among treatments. There were no differences (*p* > 0.05) among treatments for dressing percentage, rib eye area, and backfat thickness. Findings suggest that at least up to 30% LF-DDGS can be included in diets of castrated male goats without affecting production performance and carcass characteristics.

## 1. Introduction

Ethanol is a renewable fuel that is widely used by blending in gasoline to reduce the emission from vehicles, and distillers dried grains with solubles (DDGS) is a coproduct produced alongside ethanol, with high nutritional value for animals. United States (US) is the leader in bioethanol production, which alone produced 15 billion gallons of ethanol in 2021, approximately 55% of the world’s ethanol production [1]. Along with the increase in production of ethanol over last decade, production of its coproduct, DDGS, is also increasing; 22.2 million tons of DDGS were produced in 2021 in US alone [2]. In US, DDGS are predominately low fat, as the economics favor separating as much oil as possible for sale as renewable diesel feedstock and also as an ingredient in swine and poultry feed.

Low-fat distillers dried grain with solubles (LF-DDGS) is produced after distillers corn oil is extracted by heating and centrifuging thin stillage after it has been separated from whole stillage. In US, 94% of total ethanol production is from corn [3], where over 90% of the entire bioethanol plants in US have been using various facilities for oil extraction [4]. In 2021, nearly 2 million tons of corn distillers oil (CDO) was produced in US. Thus, the production of low-fat distillers dried grains with solubles is expected to increase.

DDGS is a nutritive feedstuff for livestock [5,6,7], along with poultry [8,9,10] and fish [11]. Nutrient content of DDGS varies according to feedstocks, ethanol production process (dry or wet), and plants, and it also varies between each batch of production. Different researchers have observed different ranges of crude protein (25% to 31%), ether extract (9% to 13%), neutral detergent fiber (NDF) (25% to 53%), acid detergent fiber (ADF) (9% to 15%), and starch (0% to 8%) [7,12]. Some reports recommend DDGS up to only 20% of the total diet in lactating dairy cattle, and some suggest 40% to 50% of the total ration can also be included in the finishing cattle diet without any negative impact on performance parameters [6,11]. Gurung et al. (2009) [13] observed no effect of inclusion of DDGS up to 30% in growing meat goat diets. In another study, Gurung et al. (2012) [14] found no adverse effect on digestibility and passage kinetics by including 38.1% DDGS in the diet of castrated male goats. Similarly, Sorensen et al. (2021) [15] reported that 100% of the soybean meal (SBM) could be replaced from finishing goat diets with corn DDGS without any effect on growth performance and carcass characteristics [15]. Mjoun et al. (2010) [16] found a similar milk production and dry matter intake (DMI) of a lactating cattle-feeding soybean-based diet with up to 30% reduced fat DDGS (3.5% EE) [16]. Jacela et al. (2011) [17] reported that LF-DDGS contain higher CP but less energy content than DDGS, but when reduced fat was overcome by adding dietary fat, growth performance was not affected in nursing pig, whereas the growth performance was reduced in finishing pigs. Data regarding the usage of LF-DDGS for small ruminants, especially meat goats, are limited. Thus, this research objective was to determine the effect of LF-DDGS on the growth performance, rumen fermentation, blood metabolites, and carcass characteristics in growing meat goats.

## 2. Materials and Methods

### 2.1. Experimental Animals and Diets

This study was conducted at the Caprine Research and Education Unit at George Washington Carver Agricultural Experiment Station, Tuskegee University of Tuskegee, Alabama, USA. All experimental procedures, care, and handling of the animals were approved by the Institutional Animal Care and Use Committee of Tuskegee University. Goats were purchased from Spring Acres Farm, Quitman, Georgia, USA, and were quarantined for three weeks before starting the trial. After the quarantine period, goats were individually housed in 1.1 m × 1.2 m pens with plastic slatted floors inside the indoor barn. Each pen was fitted with individual hayracks, concentrate troughs, and automated watering nipples. Twenty-four Kiko crossbreed castrated male goats (26.7 ± 1.21 kg initial bodyweight, 5–6 months of age) were randomly assigned to one of the four experimental diets: 0%, 10%, 20%, and 30% LF-DDGS supplementation diet on an as-fed basis.

LF-DDGS used to formulate the experimental diet was Dakota Gold LF-DDGS purchased from Poet Nutrition, Inc., in Sioux Falls, South Dakota. A composite sample from both of the one-ton tote bags was collected and sent to Holmes Laboratory in Millersburg, Ohio, for a nutrient analysis. Each treatment diet consisted of Bermudagrass hay (Cynodon dactylon), cracked corn, soybean meal, LF-DDGS, liquid molasses, and goat premix mixed at varying proportions, as shown in Table 1. Proportion of soyabean meal and corn was reduced/replaced by increasing the inclusion of LF-DDGS in the experimental diets. Diets were formulated according to NRC (2007) to meet or exceed all nutritional requirements for growing meat goats and were mixed at a local feed mill (The Feed Mill, Eclectic, AL, USA).

Animals were fed their respective diets for approximately three weeks for diet acclimatization before the first data collection and then continued for 84 days. Each goat was offered 50% hay and 50% grain mixes separately, calculated as 3 percent of the individual’s body weight (BW) and ad libitum water as well as access to free-choice salt blocks. Grain mixes contained 0, 20, 40, and 60% LF-DDGS so that final diets contained 0, 10, 20, and 30% LF-DDGS on an as-fed basis. Each goat’s refused grain mixes and hay were collected, weighed, and recorded daily. The amount of feed offered was adjusted at three-day intervals; if the average weight of grain mix or hay refusal for the previous three days was less than 10% of the feed offered, then grain mix or hay offered was increased by 100 g for the next three days.

### 2.2. Sample Collection of Feed and Analysis

Composite samples of the grain mix for the respective treatments and Bermudagrass hay (BGH) were shipped to Holmes Laboratory (Millersburg, OH, USA) for analysis of the DM, CP, ADF, acid hydrolysis fat, ash, phosphorus, magnesium, potassium, sulfur, manganese, zinc, and iron. Nutritional analyses were completed according to the methods described by the American Organization of Analytical Chemists [18]. NDF concentration was determined utilizing an ANKOM 2000 fiber analyzer (Ankom Technology, Macedonia, NY, USA) according to the manufacturer’s recommendations. Lignin concentration was determined according to USDA’s (1970) Forage Fiber Analysis Handbook Procedure 397 [19].

### 2.3. Rumen Fermentation

Rumen fluid samples from goats were collected on day 84 of the trial. Samples were collected two hours after feeding to ensure adequate ingesta in the ruminal cavity at the sampling time. A speculum was placed over the tongue of the oral cavity to prevent animals from chewing and biting the tube during the process. The lubricated, beveled end of a 7 mm wide tube was slowly maneuvered through the esophagus and into the rumen. A 500 mL syringe was used as the vacuum pump to collect 30 mL of rumen fluid.

The rumen fluid’s pH was determined using a pH meter. Three milliliters of 50% hydrochloric acid was added to each sample to prevent further fermentation and then stored at −20 °C for 24 h. Rumen samples were then thawed, and approximately 2 mL of the rumen fluid was transferred to microtubes. Microtubes were centrifuged at 4000 rfc for 20 min. Samples were then prepared according to the method described by Erwin et al. (1961) [20] using 25% metaphosphoric acid and ethyl butyrate as internal control; concentrations of acetate, propionate, and butyrate were determined utilizing an Agilent 7890 GC (Santa Clara, CA, USA), equipped with a flame ionization detector (FID) and a DB-WAXetr capillary column (30 m × 0.25 mm × 0.25 µm; Agilent, Santa Clara, CA, USA). The flow rate of helium was 1 mL/min. The injector temperature was set at 185 °C, with an injecting volume of 1 µL and a split inlet ratio of 4:1. The temperature of the FID detector was set at 250 °C. The oven temperature was programmed from 80 °C (held for 1 min) to 200 °C (held for 15 min) at a rate of 10 °C/min. Total VFAs were calculated according to the method described by Hall et al. (2015) [21].

### 2.4. Growth Performance and Blood Metabolites

Animals’ body weights were recorded before feeding on days 0 and day 84 using Sheep and Goat Digital Scale (Lakeland Farm and Ranch Direct), and BW gain, average daily gain (ADG), and gain-to-feed ratio (G: F) were calculated. In addition, blood samples were collected on day 84 of the trial. From each animal, 10 mL of blood was collected via the jugular venipuncture technique in a blood collection tube with a serum separator and a tube with ethylenediaminetetraacetic acid (EDTA). Blood samples were then sent to the Tuskegee University Clinical Pathology Diagnostic Laboratory (Tuskegee, AL, USA) for hematological and serum biochemistry analysis. Serum chemical analysis was conducted using an IDEXX Catalyst DX system, which also provided reference ranges for each parameter (IDEXX Laboratories, Westbrook, ME, USA). Several serum chemical parameters were assessed to understand how the LF-DDGS supplementation affected organ function and overall animal health.

### 2.5. Carcass Quality

Animals were humanely slaughtered at the end of the trial according to USDA guidelines at Fort Valley State University Meat Laboratory (Fort Valley, GA, USA). Twenty-four hours after arriving at the facility, the fasting weights of the goats were recorded before slaughtering. Hot carcass weights (HCWs, kg) were measured after the removal of offal. Chilled carcass weight (CCW, kg) was recorded 24 h after storing hot carcasses in a chilled room at 4 °C. Carcass evaluations, such as longissimus muscle area (LMA) and fat depth over the midpoint of longissimus muscle at 12th rib, were estimated by a certified USDA grader based on the USDA’s (2001) [22] series criteria, and dressing percentage (DP) and carcass shrink were calculated.

### 2.6. Statistical Analysis

Data were analyzed using the general linear model procedures of SAS (SAS Institute Inc., Cary, NC, USA). Effects of varying the inclusion amounts of LF-DDGS on the DMI, VFA profiles, blood serum metabolites, and carcass characteristics were tested by orthogonal polynomial regression. In addition, an orthogonal contrast test for equal-spaced treatments was conducted to determine the linear and quadratic effects (SAS/STAT user guide, 2004) [23].

## 3. Results

### 3.1. Nutrient Composition of the Experimental Diets

Nutrient composition of the grain mixes, BGH, and LF-DDGS used in the trials are presented in Table 2. Average CP concentration of the grain mixes with different percentages of LF-DDGS were 20.14, 21.17, 19.75, and 24.68% for 0%, 20%, 40%, and 40% LF-DDGS, respectively. Crude fat content of the LF-DDGS used in this diet was 5.75%, and fat percentage increased in grain mixes with the increase in inclusion rate of LF-DDGS (3.77, 4.40, 4.63, and 4.78% for 0%, 20%, 40%, and 60% LF-DDGS, respectively). Similarly, NDF and TDN concentrations were 8.57%, 12.48%, 14.25%, and 21.88% and 87.09%, 86.49, 85.22, and 83.71% for the 0%, 20%, 40%, and 460% LF-DDGS grain mixes, respectively. Two minerals of concern with LF-DDGS, phosphorous and sulfur, were 0.53, 0.59, 0.69, 0.84, and 0.95% and 0.27, 0.41, 0.57, and 0.74% for the 0%, 10%, 20%, and 30% LF-DDGS grain mixes, respectively.

### 3.2. Dry Matter Intake, Growth Performance, and Efficiency

In this experiment, there were no differences for DMI (*p* > 0.05) among experimental groups; however, DMI tended to follow the quadratic function (*p* = 0.05) (Table 3). DMI intake was found to be 997, 1075, 1177, and 1052 g/day for goats with 0%, 10%, 20%, and 30% LF-DDGS supplementation diets, respectively. Total live weight gains observed were 10.74, 12.35, 13.48, and 9.40 kg for 0%, 10%, 20%, and 30% LF-DDGS, which were statistically not different (*p* > 0.05) among the groups. Average daily gains followed a similar pattern as the total weight gain (111.77, 128.44, 140.30, and 97.89 g/day for the 0%, 10%, 20%, and 30% LF-DDGS included diets, respectively) at *p* > 0.05. Gain-to-feed ratios were observed to be similar among experimental groups (*p* > 0.05) (0.10, 0.11, 0.11, and 0.08 for the goats fed with 0%, 10%, 20%, and 30% LF-DDGS, respectively).

### 3.3. Rumen Fermentation and pH

pH values of rumen fluid samples were 6.31, 6.32, 6.35, and 6.43 for 0%, 10%, 20%m and 30% LF-DDGS diets, respectively, and observed pHs were similar (*p* > 0.05) among experimental groups.

Observed concentrations of acetate, propionate, butyrate, and acetate:propionate ratios are presented in Figure 1. Acetate concentrations were 50.0, 49.1, 45.1, and 39.0 mmol/lit for 0%, 10%, 20%, and 30% LF-DDGS inclusion diets, respectively. Propionate concentrations were 15.3, 14.4, 14.4, and 12.2 mmol/L while values for the butyrate concentrations were 9.7, 8, 8, and 5 mmol/L, respectively. Acetate: propionate ratios (A:P) of 3.7, 3.6, 3.3, and 3.3 were observed for 0%, 10%, 20%, and 30% LF-DDGS diets, respectively. Mean concentrations of acetate, propionate, butyrate, and the acetate-to-propionate ratio (A: P ratio) were similar (*p* > 0.05) among experimental groups (Figure 1).

### 3.4. Blood Metabolites

The value of different serum metabolites are presented in Table 4. Cholesterol concentrations increased linearly (*p* = 0.004) with an increasing amount of LF-DDGS included in the diet. The cholesterol concentrations were 51.75, 58.00, 63.92, and 67.17 mg/dl for the goats fed with 0%, 10%, 20%, and 30% LF-DDGS, respectively, which were within the normal cholesterol range in goats (80–130 mg/dL) [24]. Blood urea nitrogen (BUN) level tended to increase linearly (*p* = 0.07) with increasing inclusion of LF-DDGS (14.75, 14.60, 16.75, and 17.17 mg/dL for 0%, 10%, 20%, and 30% LF-DDGS) diets, respectively.

Blood glucose concentration decreased linearly (*p* = 0.03) with increasing inclusion of LF-DDGS in the diet (65.75, 67.20, 65.0, and 55.76 mg/dL for 0%, 10%, 20%, and 30% LF-DDGS), but this was also within the normal range of 50–75 mg/dL (Merck Veterinary Manual). Concentration of creatine kinase (CK) and aspartate aminotransferase (AST) concentration increased linearly (*p* < 0.05) with an increasing amount of inclusion of LF-DDGS in the diet, and the concentration of gamma-glutamyl transferase (GGT) decreased linearly (*p* < 0.05) with an increasing amount of LF-DDGS inclusion in the diet. However, all values of CK, AST, and GGT, along with other serum biochemistry profiles (Table 4), were within the normal range as per reference values provided by College of Veterinary Medicine, Cornell University [25].

### 3.5. Carcass Characteristics

Dressing percentage, longissimus muscle area (LMA), fat depth (at 12th rib), and carcass shrink percentage (Table 5) were found to not be affected (*p* > 0.05) by inclusion of different levels of LF-DDGS in castrated goat diet. Dressing percentages of 38.55%, 40.70%, 40.60%, and 39.86%; LM areas of 13.76, 13.87, 15.32, and 12.47 cm^2^; and back fat thicknesses of 2.17, 1.92, 1.93, and 1.77 mm were observed for goats fed with 0%, 10%, 20%, and 30% LF-DDGS diets, respectively.

## 4. Discussion

### 4.1. Feed Composition

Nutrient content of hay varies with maturity of the plant harvested and method of harvest, storage, and many other factors. CP and TDN in the BGH used in this trial were greater than values provided in the Nutrient Requirement of Dairy Cattle (2001) [26] (5% CP and 52–55% TDN), whereas the NDF and ADF values were less than values in the same book (73% NDF and 36% ADF) [27]. In another report, Preston et. al. (2016) observed DM (89%), CP (10%), ADF (32%), NDF (72%), and TDN (53%) [28]. Difference might be due to growing conditions and harvesting time. Nutrient composition of ethanol coproducts varied with raw materials’ factors, such as grain type, grain variety, and grain quality, as well as processing factors, such as grind procedure, fineness, and duration [27,29,30]. However, nutrient composition of LF-DDGS used in this trial was similar to the LF-DDGS reported by other authors [17,31]. Even though all of the experimental diets were formulated to be iso-nitrogenous, greater CP in the diet with 30% LF-DDGS might be because we used standard book values for other ingredients while calculating the nutrient composition of grain mixes.

Increase in ADF and NDF concentrations with increase in inclusion of a percentage of LF-DDGS in grain mixes was expected due to 9.71% ADF and 25.28% NDF in LF-DDGS used to formulate diets. Non-fiber carbohydrate (NFC) and TDN values and net energy for growth decreased in grain mixes with inclusion rate of LF-DDGS, which were also expected because of the decreased NFC (33.80%) in LF-DDGS that was used in this trial and increasing amount of ADF with an increase in the inclusion rate of LF-DDGS.

NRC (2007) [32] recommends 5% crude fat for growing goats, and greater amount of fat in the diet will inhibit rumen function in ruminants [33]. One of the concerns of feeding DDGS is increased fat content (9–13%). In LF-DDGS, as the fat is reduced due to oil extraction, fat content becomes suitable for a ruminant diet. In this experiment too, fat content of all of diets remained less than 5%, even though fat content increased with an increase in LF-DDGS inclusion (Table 2). Other researchers also noted a similar increase in fat percentage with increase in the percentage of reduced-fat DDGS (Mjoun et al., 2010; Nelson, Hohertz, DiCostanzo, & Cox, 2014). This might be because of the lower crude fat content in the other ingredients included in the formulated grain mixes used in this experiment.

Another concern with feeding DDGS is greater concentration of phosphorous and sulfur [34]. However, having increased phosphorus in LF-DDGS can decrease the cost of animal feeds, because phosphate is a relatively high-cost ingredient. The recommended phosphorous concentration in a goat diet is from 0.14 to 0.25% of the diet (on a DM basis) (NRC 2007) [32], and the sulfur content should not be more than 0.3% of the total diet (on a DM basis) [34]. General recommendation for the calcium:phosphorous ratio for livestock is 1:1 to 2:1. But phosphorous content was greater than calcium in grain mixes used in this experiment. BGH (which was 50% of total diet in this experiment) contained 0.46% Ca and 0.19% P, which reduced the percentage of P in total diet. However, there was still a greater P content than Ca in the total diet with 20% and 30% LF-DDGS. Thus, it is recommended to adjust the Ca:P ratio by supplemental Ca while formulating the grain mixes.

In a review by Drewnoski et al. (2014) [35], authors noted that sulfur-induced polio-encephalomalacia (S-PEM) is a common risk with a DDGS ration in cattle. However, the risk of S-PEM can be reduced by including a minimum of 7–8% NDF (on a DM basis) in the diet with 0.4% sulfur or more [35]. In this study, the sulfur content in the grain mixes were more than recommended. However, no symptoms of S-PEM or other unusual symptoms were noticed, which could be because of a reduction in the sulfur percentage due to inclusion of 50% BGH and also due to the presence of more NDF (42.06%) in BGH.

An additional concern for feeding any byproduct including LF-DDGS is the mycotoxin content. Mycotoxin content in LF-DDGS used in this experiment was not evaluated, which is recommended in further research.

### 4.2. Dry Matter Intake, Growth Performance, and Efficiency

There are no published results available on DMI using LF-DDGS in goats. Thus, comparisons were made with other ruminant species and traditional DDGS (high-fat DDGS) in ruminant species, including goats. Therefore, results presented in current study should be treated with caution.

Similar to our research findings, Mjoun et al. (2010) [16] found no difference in DMI in mid-lactating cows when reduced-fat DDGS was included in diet at 0%, 10%, 20%, and 30% on a DM basis [16]. Nelson et al. (2010) [36] also found no difference for DMI when reduced-fat DDGS were incorporated at 0%, 15%, 30%, and 45% of diet. In contrast, Testroet et al. (2018) reported an increased DMI (*p* < 0.01) in lactating Holstein dairy cows when supplied with 20% reduced-fat DDGS than control diet with no reduced-fat DDGS. Similarly, Ramirez-Ramirez et al. (2016) [37] also reported an increase in DMI (*p* < 0.01) when 30% DDGS or 30% reduced-fat DDGS was supplemented to lactating Holstein cows in comparison to no DDGS supplementation in the diet. In an experiment by Gurung et al. (2009) [13], DMI of growing Kiko × Spanish male goats supplied with 0%, 10%, 20%, and 30% DDGS were similar (*p* = 0.62). Similarly, in a recent paper by Dahmer et al. (2022) [38], no difference was found for DMI of Boer goats when supplied with 33% DDGS compared to 0% DDGS. Sorensen et al. (2021) [15] also reported no difference in the DMI when 0%, 33%, 66%, or 100% soybean meal was replaced by DDGS in finishing rations of Boer cross goats.

In contrast to this research, Dahmer et al. (2022) [38] found a decrease in average daily gain when 33% DDGS was included in the diet of Boer kids for 21 days compared with 0% DDGS. However, Sorensen et al. (2021) [15] found a linear increase in the average daily gain (0.19, 0.22, 0.22, and 0.28 g/day for 0%, 10%, 20%, and 30% corn DDGS) with an increased DDGS inclusion in Boer cross goats of age 70 days for a trial for 47 days. In contrast with findings of this research, Sorensen et al. (2021) [15] observed a difference (*p* < 0.05) in the gain-to-feed ratio when Boer goat kids were fed 0% or 10% DDGS (0.18 and 0.19 for 0% and 10%, respectively) in comparison to kids fed with 20% or 30% DDGS (0.22 and 0.28 for 20% and 30%, respectively).

### 4.3. Rumen Fermentation and Blood Metabolites

Volatile fatty acids are the major energy source for ruminants including goats. Similar to our findings, Benchaar et al. (2013) [39] found a similar decrease in acetate concentration with an increasing amount of DDGS (0, 10, 20, and 30% inclusion on a DM basis) fed to dairy cows. A decrease in concentration of propionic acid was expected because of decreasing nonfiber carbohydrate (NFC) with an increasing amount of LF-DDGS. However, Benchaar et al. (2013) [39] reported a linear increase in propionic and butyric acid with an increased concentration of DDGS in diets of dairy cows (*p* < 0.05). The optimal A:P ratio in ruminants is more than 2.2 to 1 [40]. In this trial, the A:P ratio was greater than three in all of the experimental groups.

Feed ingredients should not affect health of animals, and the blood biochemistry can reflect the overall health of animals. All blood metabolites in the serum of goats in all of the experimental groups in this study were within the normal range, even though some of them tended to follow a linear increase or decrease with an increased LF-DDGS concentration in their diet. In agreement with results of this study, Gurung et al. (2009) [13] reported an increasing concentration of cholesterol at 41.2, 66.2, 76.5, and 83.2 mg/dL when DDGS was fed to the Kiko x Spanish male goats at 0%, 10%, 20%, and 30% inclusion rates, respectively. An increase in the cholesterol concentration could be attributable to increased fat concentration with increase in experimental diets. However, a narrow range of increase in serum cholesterol concentration in this research in comparison to what was reported by Gurung et al. (2009) [13] might be due to the use of LF-DDGS with less fat (5.75%) in this experiment rather than the higher fat DDGS (11.93%) by Gurung et al. (2009) [13].

Hammond (1983) [41] reviewed that one of the factors for an increase in serum BUN concentration is increase in dietary nitrogen intake. Linear increase in BUN with a narrow range in this experiment might be due to slight increase in protein concentration (Table 2) in experimental diet with an increasing amount of LF-DDGS inclusion. Gurung et al. (2009) [13] reported no difference in BUN concentration when DDGS was fed to Kiko x Spanish male goats at 0%, 10%, 20%, and 30% inclusion rates.

Increased starch in the diet increased the serum glucose concentration of ruminants [42]. Thus, observed decrease in the serum glucose concentration in this experiment might be due to the decrease in starch in diet as LF-DDGS was increasingly replacing corn in experimental diets.

### 4.4. Carcass Characteristics

Findings of this research regarding carcass traits were in agreement with findings by Dahmer et al. (2022) [38], where they found no difference in dressing percentage, LMA, and fat thickness when Boer goats were fed with either 0% DDGS or 33% DDGS diets for 21 days. Gurung et al. (2009) [13] also reported similar findings in Kiko x Spanish male goats of 4–5 months of age fed with 0%, 10%, 20%, and 30% DDGS diets. Furthermore, in agreement with these research findings, Sorensen et al. (2021) [15] also reported no effect on the carcass traits in Boer cross goats fed with 0%, 10%, 20%, and 30% DDGS levels. However, dressing percentage of goats observed (39–41%) in this experiment were less than reported by Dahmer et al. (2022) [38] (47–48%), Gurung et al. (2009) [13] (42–45%), and Sorensen et al. (2020) [15] (49–50%) in male goats. Factors such as age, slaughter weight, diet, castration, sex, genotype, and gut fill influence goat dressing percentage [43]. Plane of nutrition and gut fill might be reasons for this experiment’s low DP.

Observed LMA was similar to LMA reported by Sorensen et al. (2020) [15] (11.60, 13.10, 13.30, and 13.50 cm^2^ for the 0, 10, 20, and 30% DDGS amounts, respectively) in Boer cross male goats. However, observed LMA values were greater than those reported by Gurung et al. (2009) [13] at 9.75, 10.25, 9.50, and 9 cm^2^ in Kiko x Spanish goats fed with 0%, 10%, 20%, and 30% DDGS-containing diets.

In a trial by Dahmer et al. (2022) [38], back fat thickness was only 1.1 mm for Boer goats fed 0% DDGS and 1.0 mm for goats fed 33% DDGS, which is less than the back fat thickness observed in this experiment. This might be because fat content in the diet provided to goats by Dahmer et al. (2022) [38] was 2–3%, whereas fat content in this experiment was 4–5%.

## 5. Conclusions

Results suggest that at least up to 30% LF-DDGS can be included in a male goat diet, completely replacing soybean meal and partly replacing corn without affecting production performance and carcass characteristics. At an LF-DDGS concentration greater than 10% of the total diet, the Ca:P ratio should be adjusted by additional Ca supplements. LF-DDGS has the potential to be a sustainable alternative protein source for meat goats as long as it is competitively priced. Further research is also warranted to determine the effect of inclusion of LF-DDGS higher than 30% as well as its economic inclusion rate.

## Figures and Tables

**Figure 1 animals-12-03318-f001:**
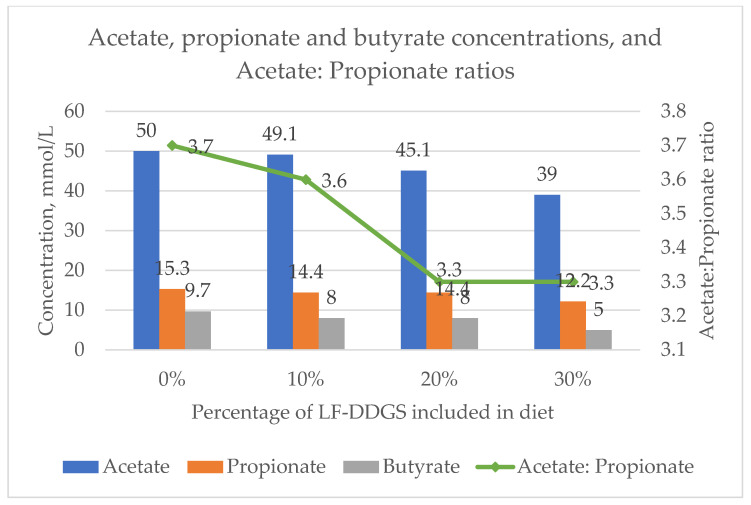
Acetate, propionate, and butyrate concentrations and the acetate:propionate ratio of castrated male goats fed with different levels of low-fat distillers dried grains with solubles (LF-DDGS) for 84 days.

**Table 1 animals-12-03318-t001:** Composition of experimental diets containing varying levels of low-fat distillers dried grains with solubles (LF-DDGS) and hay used for 84 day feeding period (on an as-fed basis).

Ingredient, % of Diet	Percentage of Low-Fat DDGS in the Diet, %			
	0	10	20	30
Bermudagrass Hay	50	50	50	50
Cracked Corn	34	28.5	23	16.5
Soybean Meal 48%	12.5	8	3.5	0
Low-Fat DDGS	0	10	20	30
Liquid Molasses	2.5	2.5	2.5	2.5
Goat Premix ^†^	1	1	1	1
Total	100	100	100	100

LF-DDGS = low-fat distillers dried grains with solubles. ^†^ Purina^®^ Goat Mineral with a minimum of 15.3% calcium and 8% phosphorous; 50 ppm selenium; 4000 ppm zinc; 300,000 IU/lb vitamin A.

**Table 2 animals-12-03318-t002:** Analyzed chemical composition of grain mixes, Bermudagrass hay (BGH), and low-fat distillers dried grains with solubles (LF-DDGS) used in experimental diets fed over 84 days to growing castrated male goats.

Nutrient Analysis ^†^	Percentage of LF-DDGS in Grain Mixes, %				BGH	LF-DDGS
	0	20	40	60		
Moisture, %	13.68	12.87	13.07	13.53	14.25	12.55
Dry Matter (DM), %	86.32	87.13	86.93	86.47	85.75	87.45
Crude Protein, %	20.14	21.17	19.75	24.68	9.28	29.95
Available Protein, %	19.88	20.77	19.36	24.08	-	-
Adjusted Crude Protein, %	20.14	21.17	19.75	24.68	-	-
ADF Protein, %	0.26	0.4	0.39	0.6	-	-
NDF Protein, %	0.49	0.72	0.74	1.1	-	-
Lignin, %	0.21	0.21	0.21	0.21	-	-
Acid Detergent Fiber, %	2.93	3.76	4.17	7.29	26.80	9.71
Neutral Detergent Fiber, %	8.57	12.48	14.25	21.88	42.06	25.28
Non-Fiber Carbohydrate (NFC), %	62.42	56.77	55.11	43.01	-	33.80
Crude Fat, %	3.77	4.4	4.63	4.78	-	5.75
TDN, %	87.09	86.49	85.22	83.71	66.37	84.59
NEl, Mcal/kg	2.01	2.00	1.97	1.93	0.68	1.90
NEm, Mcal/kg	2.15	2.13	2.10	2.05	0.69	2.07
NEg, Mcal/kg	1.48	1.46	1.43	1.39	0.42	1.41
Ash, %	4.32	4.74	5.88	5.87	-	5.24
Lignin Insoluble Ash, %	0.21	0.21	0.21	0.21	-	-
Calcium (Ca), %	0.25	0.31	0.27	0.32	0.46	0.05
Phosphorus (P), %	0.53	0.59	0.69	0.84	0.19	0.95
Magnesium (Mg), %	0.23	0.19	0.21	0.27	0.18	0.26
Potassium(K), %	1.26	1.28	1.28	1.31	1.56	0.92
Sulfur (S), %	0.27	0.41	0.57	0.74	-	0.083
Sodium (Na), %	0.314	0.376	0.477	0.45	0.07	-
Copper (Cu), ppm	26	32	34	31	3	8
Manganese (Mn), ppm	52	57	62	54	67	-
Zinc (Zn), ppm	106	161	165	156	19	51
Iron (Fe), ppm	143	190	182	172	28	-
Nitrate (NO^3^)	Negative	Negative	Negative	Negative	-	-

^†^ All values are on a dry matter basis except for moisture and DM. ADF = acid detergent fiber; NDF = neutral detergent fiber; TDN = total digestible nutrient; NEl = net energy for lactation; NEm = net energy for maintenance; Neg = net energy for growth; Mcal/kg = megacalorie per kilogram.

**Table 3 animals-12-03318-t003:** Dry matter intake and growth performance of castrated male goats fed diets with different percentages of low-fat distillers dried grains with solubles (LF-DDGS) for 84 days.

Parameters	Percentage of LF-DDGS in Diets, %					*p*-Value ^†^	
	0%	10%	20%	30%	SEM	Linear	Quadratic
DMI, g/day	997	1075	1177	1052	19.60	0.21	0.05
Initial BW, kg	28.46	27.24	29.97	27.61	1.01	0.99	0.82
Final BW, kg	39.21	39.58	43.45	37.02	0.94	0.79	0.16
Total live weight gain, kg	10.74	12.35	13.48	9.40	0.60	0.66	0.07
Average daily gain, g/day	111.77	128.44	140.30	97.89	6.26	0.66	0.49
Gain:Feed Ratio	0.10	0.11	0.11	0.08	0.004	0.30	0.71

^†^ Based on the orthogonal contrast for equally spaced treatments (n = 6 animals per experimental group). SEM = standard error of mean.

**Table 4 animals-12-03318-t004:** Select-serum biochemistry profile of the castrated male goats fed diets with different percentages of low-fat distillers dried grains with solubles (LF-DDGS) for 84 days.

Serum Chemistry	Percentage of LF-DDGS in Diets, %					*p*-Values ^†^	
	0	10	20	30	SEM	Linear	Quadratic
Cholesterol, mg/dL	51.75	58.00	63.92	67.17	1.49	0.004 **	0.69
Creatine Kinase, U/L	221.58	275.60	287	294	8.35	0.02 *	0.27
Alanine Aminotransferase, U/L	4.67	7.50	4.41	4.67	0.35	0.42	0.15
Amylase, U/L	78.08	43.20	29.92	39.75	8.14	0.16	0.28
Alkaline Phosphatase, U/L	409.50	993.80	1127.50	1018.33	213.53	0.40	0.52
Total Protein, g/dL	10.02	6.48	7.05	6.62	0.80	0.28	0.44
Glucose, mg/dL	65.75	67.20	65.00	55.76	1.27	0.03 *	0.10
Phosphorus, mg/dL	12.67	9.42	9.13	10.35	1.01	0.51	0.38
Bilirubin, Total, mg/dL	0.13	0.13	0.19	0.17	0.01	0.15	0.05
Blood Urea Nitrogen, mg/dL	14.75	14.60	16.75	17.17	0.46	0.07	0.80
Creatinine, mg/dL	0.64	0.61	0.58	0.63	0.01	0.57	0.24
Carbon Dioxide, mmol/L	22.53	22.29	22.78	20.26	0.31	0.08	0.16
Sodium, mmol/L	142.71	142.82	143.05	143.70	0.17	0.10	0.53
Potassium, mmol/L	4.75	4.87	4.99	4.99	0.04	0.07	0.56
Chloride, mmol/L	101.65	109.31	108.88	113.25	1.79	0.09	0.71
Calcium, mg/dL	9.65	9.48	9.78	9.07	0.08	0.10	0.17
Albumin, g/dL	2.24	2.50	2.60	2.41	0.03	NS	NS
Triglycerides, mg/dL	22.42	22.70	24.33	26.50	0.93	0.18	0.68
Gamma Glutamyl Transferase, U/L	37.58	36.00	28.42	32.67	0.94	0.04 *	0.21
Aspartate Aminotransferase, U/L	51.00	87.60	64.83	99.08	3.01	0.001 **	0.88
Bilirubin, Direct	0.13	0.14	0.14	0.15	0.01	0.36	0.86

^†^ Based on the orthogonal contrast for equally spaced treatments (n = 6 animals per experimental group). SEM = standard error of the mean. * Significant; ** highly significant. Serum chemistry analyzed by the IDEXX Catalyst DX system.

**Table 5 animals-12-03318-t005:** Carcass characteristics of the castrated male goats fed diets with different percentages of low-fat distillers dried grains with Solubles (LF-DDGS) for 84 days.

Carcass Trait	Percentage of LF-DDGS in Diets, %				*p*-Value ^†^		
	0	10	20	30	SEM	Linear	Quadratic
Final wt., kg	39.21	39.58	43.45	37.02	0.94	0.79	0.16
Fasting wt., kg	35.98	35.45	40.45	35.08	1.05	0.88	0.22
Hot Carcass wt., kg	21.97	20.98	23.94	20.98	0.39	0.93	0.25
Carcass Chilled wt., kg	14.92	14.35	16.64	14.10	0.40	0.90	0.26
Carcass Shrink Percentage	32.42	31.13	30.86	33.25	0.76	0.79	0.34
Dressing Percentage	38.55	40.70	40.60	39.86	0.63	0.58	0.36
Longissimus Muscle Area, cm^2^	13.76	13.87	15.32	12.47	0.36	0.43	0.93
Fat Depth at the 12th Rib, mm	2.17	1.92	1.93	1.77	0.06	0.39	0.21

^†^ Based on the orthogonal contrast for equally spaced treatments (n = 6 animals per experimental group). SEM = standard error of the mean.

## Data Availability

Data supporting the findings of this study are available to anyone from the corresponding author upon reasonable request.

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
