# Peer review of "Effects of Low-Fat Distillers Dried Grains with Solubles Supplementation on Growth Performance, Rumen Fermentation, Blood Metabolites, and Carcass Characteristics of Kiko Crossbred Wether Goats"

_animals, 2022, doi:10.3390/ani12233318_

Round 1
Reviewer 1 Report
General comments:
The objectives are sound and methodology lines up with the objectives. Results are presented and discussed with conclusions supported by the data, with an occasional “stretch” for a conclusion. The word “level(s)” is used extensively throughout the paper, and in most cases should be “concentration” or “amount” rather than “level”. Also “the” is used much more than needed, suggest removing “the” when either not needed or inappropriate (“the” is singular and is used for multiple items). Finally “significantly” should not be used in a sentence when differences (or lack of) are discussed – the P-value should be reported and the reader can determine “significance” or not.
Specific comments:
Line Comment
14 “in goat” should be “using goats”
14 “levels” should be “amounts”
17 “levels” should be “amounts”
17 “the diets” should be just “diets”
42 “tons” implies 2000 pounds NOT 1000 kg – if this is metric ton, “tonne” or “metric ton”
49 “tons” implies 2000 pounds NOT 1000 kg – if this is metric ton, “tonne” or “metric ton”
55 “different researcher” should be “different researchers”
58 “the lactating cattle” should be “lactating cattle” suggest adding “beef” or “dairy” in front of cattle as this alters the intensity of the statement
61 “diet” should be “diets”
64 “the finishing goat diet by the corn” should be “finishing goat diets with corn”
66 “in a lactating” should be “of a lactating”
82 remove “(TUACUC)” as it is not used elsewhere
84 “salted floors” – either this should be “slated floors” or an explanation is needed
92 “tons” implies 2000 pounds NOT 1000 kg – if this is metric ton, “tonne” or “metric ton”
93 “diet consists of” should be “diet consisted of”
96 “meet and exceed” should be “meet or exceed”
104 I think “mixes contained 20, 40, and 30%” should be “mixes contained 20, 40, and 60%”
108 “lower than 10” should be “less than 10”
108 intake should always be less than 10% of the BW. What do you mean with this statement? Should it be 1.0% BW?
125 “tube is” should be “tube was”
127 suggest re-write first sentence to “Rumen fluid pH was determined using a pH meter”
129 “The rumen” should be “Rumen”
130 “The microtubes” should be “Microtubes”
131 centrifuged at 6000 rpm is ambiguous as amount of g-force is different at the same rpm for different radius centrifuges – report g-force centrifuged rather than rpm
133 “the concentrations” should be “concentrations”
133 “previously described VFAs” these are NOT previously described – add this section
155 “the USDA” should be “USDA”
157 “the fasting” should be “fasting”
159 “the offal” should be “offal”
160 “the hot” should be “hot”
166 “levels of” should be “amount of”
172 “the LF-DDGS” should be “LF-DDGS”
189 remove “significant”
189 “in DMI” should be “for DMI”
190 “the DMI” should be “DMI”
191 “DMI intake” is redundant – should be just “DMI”
191 “the goats” should be “goats”
192 “the total” should be “total”
192-194 talking about “numerical differences”. There are ALWYAS numerical differences – as stated later these are NOT statistically different therefore they are NOT DIFFERENT – if the authors wish to argue they believe the statistics are wrong, then they need to justify that reasoning. If they can not do that – this “numerical” discussion should be removed
196 remove “statistically non-significant” and add “not different with the appropriate P-value.
196 “The average” should be “Average”
199 “between” should be “among”
205 “the observed” should be “observed”
206 “the experimental” should be “experimental”
207 “The observed” should be “Observed”
213-215 talking about “numerical differences”. There are ALWYAS numerical differences – as stated later these are NOT statistically different therefore they are NOT DIFFERENT – if the authors wish to argue they believe the statistics are wrong, then they need to justify that reasoning. If they can not do that – this “numerical” discussion should be removed
221 “the different” should be “different”
222 “Cholesterol levels” should be “Cholesterol concentrations”
222 “increasing levels of” should be “increasing amounts of”
223 “cholesterol levels were” should be “cholesterol concentrations were”
225 “The blood” should be “Blood”
229 “The blood” should be “Blood”
229 “glucose level” should be “glucose concentration”
232 “The level of creatine” should be “Concentration of creatine”
232 “AST level” “AST concentration”
233 “increasing level of” should be increasing amount of”
233 “the level of” should be “concentration of”
234 “was decreasing “ should be “decreased”
234 “with decreasing level of LF-DDGS” should be “with increasing amount of LF-DDGS”
239 “The dressing” should be “Dressing”
240 “to be not affected” should be “to not be affected”
252 “were higher than the values” should be “were greater than values”
253 “the NDF” should be “NDF”
253 “are lower than” should be “are less than”
255 “the DM” should be “DM”
255 “but the TDN” should be “but TDN”
256 “values were higher” should be “values were greater”
260 “the LF-DDGS” should be “LF-DDGS”
260 “the LF-DDGS” should be “LF-DDGS”
262 “the higher CP” should be “the greater CP”
263 “the nutrient” should be “nutrient”
265 “The increase in ADF” should be “Increased ADF”
267 “The Non-Fiber” should be “Non-Fiber”
269 “of lower NFC” should be “of decreased NFC”
270 “level of ADF” should be “ “amount of ADF”
271 “and the higher amount of fat” should be “and the greater fat”
272 “inhibit the rumen” should be “inhibit rumen”
273 “the higher fat” should be “the increased fat”
273 remove “on it” at the end of the sentence
274 “the fat content” should be “fat content”
276 “The other” should be “Other”
277 “with the increase in the percentage” should be “with increase percentage”
279 “the grain mixes” should be “grain mixes”
281 “the higher level of” should be “greater concentration of”
282 “higher phosphorus” should be “increased phosphorus”
282 “can lower the cost” should be “decrease cost”
284 “level in the goat” should be concentration in the goat”
285 “the sulfur” should be “sulfur”
289 “the sulfur” should be “sulfur”
289 “the grain” should be “grain”
292 “do to high NDF” should be “due to more NDF”
293 “the LF-DDGS” should be “LF-DDGS”
295 “the comparisons” should be “comparisons”
296 “the traditional” should be “traditional”
299 remove “significant”
300 “in DMI in the mid-lactating” should be “for DMI in mid-lactating”
302 “difference in the DMI when the reduced fat” should be “difference for DMI when reduced fat”
303 “a higher DMI (p < 0.01) in the lactating” should be “an increased DMI (p < 0.01) for lactating”
306 “a significant increase in DMI” should be “an increase of DMI”
307 "the lactating” should be “lactating”
308 “the DMI” should be “DMI”
310 “no statistical difference” should be “no difference”
311 “found in the DMI” should be “found for DMI”
311 “as compared” should be “compared”
313 “the finishing ration of Boer cross goat” should be “finishing rations of Boer cross goats”
318 “the increase in DDGS” should be “increased DDGS”
319 remove “significant”
326 “increasing level of” should be increasing amount of”
328 “increasing level of” should be increasing amount of”
330 “Increasing concentration” should be “increased concentration”
330 “in the diet of” should be “in diets of”
331 “was higher than” should be “was greater than”
333 “Any of the feed” should be “Feed”
337 “increase in” should be “increased”
338 “level of” should be “concentration of”
340 “cholesterol level” should be “cholesterol concentration”
342 “cholesterol level” should be “cholesterol concentration”
344 “of lower fat” should be “of less fat”
344 “than higher fat” should be “than more fat”
348 “protein level” should be “protein concentration”
349 “level of” should be “amount of”
350 remove “significant”
352 “the serum” should be “serum”
352 “in ruminants” should be “of ruminants
353 “glucose level” should be “glucose concentration”
358 remove “significant”
363 remove “significant”
371 “DDGS levels” should be “DDGS amount”
372 “were higher” should be “were greater”
375 “the back fat” should be “back fat”
376 “fed with 0%” should be fed 0%”
376 “fed with 33” should be fed 33%”
376 “than the back” should be “than back”
383 “the production” should be “production”
383 “the male” should be “male”
384 “the prices” should be “prices”
Author Response
Thank you for valuable comments regarding the submitted manuscript. We have completed the revision based on your comments. We are very grateful for these improvements to the presentation of our work. We hope that the clarification below and within the text of the submission have completed all the necessary steps to achieve acceptance for publication.
General comments:
The objectives are sound and methodology lines up with the objectives. Results are presented and discussed with conclusions supported by the data, with an occasional “stretch” for a conclusion. The word “level(s)” is used extensively throughout the paper, and in most cases should be “concentration” or “amount” rather than “level”. Also “the” is used much more than needed, suggest removing “the” when either not needed or inappropriate (“the” is singular and is used for multiple items). Finally, “significantly” should not be used in a sentence when differences (or lack of) are discussed – the P-value should be reported and the reader can determine “significance” or not.
Response to general comments:
Words “level(s)” and “the” has been replaced/removed. The use of word “significantly” has been corrected as per the reviewer’s comments.
Specific comments:
Point 1 Line: 14 “in goat” should be “using goats”
Response 1: It is corrected as per the reviewer’s comment.
Point 2 Line: 14 “levels” should be “amounts”
Response 2: It is corrected as per the reviewer’s comment.
Point 3 Line: 17 “levels” should be “amounts”
Response 3: Revised as per the reviewer’s comment.
Point 4 Line: 17 “the diets” should be just “diets”
Response 4: It is corrected as per the reviewer’s comment.
Point 5 Line: 42 “tons” implies 2000 pounds NOT 1000 kg – if this is metric ton, “tonne” or “metric ton”
Response 5: It is tons (2000 pounds)
Point 6 Line: 49 “tons” implies 2000 pounds NOT 1000 kg – if this is metric ton, “tonne” or “metric ton”
Response 6: It is tons (2000 pounds)
Point 7 Line: 55 “different researcher” should be “different researchers”
Response 7: Revised as per the reviewer’s comment.
Point 8 Line: 58 “the lactating cattle” should be “lactating cattle” suggest adding “beef” or “dairy” in front of cattle as this alters the intensity of the statement
Response 8: It is corrected as per the reviewer’s comment.
Point 9 Line: 61 “diet” should be “diets”
Response 9: Revised as per the reviewer’s comment.
Point 10 Line: 64 “the finishing goat diet by the corn” should be “finishing goat diets with corn”
Response 10: It is corrected as per the reviewer’s comment.
Point 11 Line: 66 “in a lactating” should be “of a lactating”
Response 11: It is corrected as per the reviewer’s comment.
Point 12 Line: 82 remove “(TUACUC)” as it is not used elsewhere
Response 12: It is corrected as per the reviewer’s comment.
Point 13 Line: 84 “salted floors” – either this should be “slated floors” or an explanation is needed
Response 13: It is corrected as per the reviewer’s comment.
Point 14 Line: 92 “tons” implies 2000 pounds NOT 1000 kg – if this is metric ton, “tonne” or “metric ton”
Response 14: It is tons (2000 pounds)
Point 15 Line: 93 “diet consists of” should be “diet consisted of”
Response 15: It is corrected as per the reviewer’s comment.
Point 16 Line: 96 “meet and exceed” should be “meet or exceed”
Response 16: It is corrected as per the reviewer’s comment.
Point 17 Line: 104 I think “mixes contained 20, 40, and 30%” should be “mixes contained 20, 40, and 60%”
Response 17: It is corrected as per the reviewer’s comment.
Point 18 Line: 108 “lower than 10” should be “less than 10”
Response 18: It is corrected as per the reviewer’s comment.
Point 19 Line: 108 intakes should always be less than 10% of the BW. What do you mean with this statement? Should it be 1.0% BW?
Response 19: It is corrected as “10% of feed offered”.
Point 20 Line: 125 “tube is” should be “tube was”
Response 20: It is corrected as per the reviewer’s comment.
Point 21 Line: 127 suggest re-write first sentence to “Rumen fluid pH was determined using a pH meter”
Response 21: It is corrected as per the reviewer’s comment.
Point 22 Line: 129 “The rumen” should be “Rumen”
Response 22: It is corrected as per the reviewer’s comment.
Point 23 Line: 130 “The microtubes” should be “Microtubes”
Response 23: Revised as per the reviewer’s comment.
Point 24 Line: 131 centrifuged at 6000 rpm is ambiguous as amount of g-force is different at the same rpm for different radius centrifuges – report g-force centrifuged rather than rpm
Response 24: It is revised to g-force.
Point 25 Line: 133 “the concentrations” should be “concentrations”
Response 25: It is corrected as per the reviewer’s comment.
Point 26 Line: 133 “previously described VFAs” these are NOT previously described – add this section
Response 26: It is revised as per the reviewer’s comment.
Point 27 Line: 155 “the USDA” should be “USDA”
Response 27: It is corrected as per the reviewer’s comment.
Point 28 Line: 157 “the fasting” should be “fasting”
Response 28: It is corrected as per the reviewer’s comment.
Point 29 Line: 159 “the offal” should be “offal”
Response 29: It is corrected as per the reviewer’s comment.
Point 30 Line: 160 “the hot” should be “hot”
Response 30: It is corrected as per the reviewer’s comment.
Point 31 Line: 166 “levels of” should be “amount of”
Response 31: It is revised as per the reviewer’s comment.
Point 32 Line: 172 “the LF-DDGS” should be “LF-DDGS”
Response 32: It is corrected as per the reviewer’s comment.
Point 33 Line: 189 remove “significant”
Response 33: It is corrected as per the reviewer’s comment.
Point 34 Line: 189 “in DMI” should be “for DMI”
Response 34: It is revised as per the reviewer’s comment.
Point 35 Line: 190 “the DMI” should be “DMI”
Response 35: It is corrected as per the reviewer’s comment.
Point 36 Line: 191 “DMI intake” is redundant – should be just “DMI”
Response 36: It is corrected as per the reviewer’s comment.
Point 37 Line: 191 “the goats” should be “goats”
Response 37: It is corrected as per the reviewer’s comment.
Point 38 Line: 192 “the total” should be “total”
Response 38: It is corrected as per the reviewer’s comment.
Point 39 Line: 192-194 talking about “numerical differences”. There are ALWYAS numerical differences – as stated later these are NOT statistically different therefore they are NOT DIFFERENT – if the authors wish to argue they believe the statistics are wrong, then they need to justify that reasoning. If they cannot do that – this “numerical” discussion should be removed
Response 39: “Numerical” discussion has been removed.
Point 40 Line: 196 remove “statistically non-significant” and add “not different with the appropriate P-value.
Response 40: It is corrected as per the reviewer’s comment.
Point 41 Line: 196 “The average” should be “Average”
Response 41: It is revised as per the reviewer’s comment.
Point 42 Line: 199 “between” should be “among”
Response 42: It is corrected as per the reviewer’s comment.
Point 43 Line: 205 “the observed” should be “observed”
Response 43: It is corrected as per the reviewer’s comment.
Point 44 Line: 206 “the experimental” should be “experimental”
Response 44: It is corrected as per the reviewer’s comment.
Point 45 Line: 207 “The observed” should be “Observed”
Response 45: It is corrected as per the reviewer’s comment.
Point 46 Line: 213-215 talking about “numerical differences”. There are ALWYAS numerical differences – as stated later these are NOT statistically different therefore they are NOT DIFFERENT – if the authors wish to argue they believe the statistics are wrong, then they need to justify that reasoning. If they cannot do that – this “numerical” discussion should be removed
Response 46: It is corrected as per the reviewer’s comment.
Point 47 Line: 221 “the different” should be “different”
Response 47: It is corrected as per the reviewer’s comment.
Point 48 Line: 222 “Cholesterol levels” should be “Cholesterol concentrations”
Response 48: It is corrected as per the reviewer’s comment.
Point 49 Line: 222 “increasing levels of” should be “increasing amounts of”
Response 49: It is revised as per the reviewer’s comment.
Point 50 Line: 223 “cholesterol levels were” should be “cholesterol concentrations were”
Response 50: It is corrected as per the reviewer’s comment.
Point 51 Line: 225 “The blood” should be “Blood”
Response 51: It is corrected as per the reviewer’s comment.
Point 52 Line: 229 “The blood” should be “Blood”
Response 52: It is corrected as per the reviewer’s comment.
Point 53 Line: 229 “glucose level” should be “glucose concentration”
Response 53: It is corrected as per the reviewer’s comment.
Point 54 Line: 232 “The level of creatine” should be “Concentration of creatine”
Response 54: It is corrected as per the reviewer’s comment.
Point 55 Line: 232 “AST level” “AST concentration”
Response 55: It is corrected as per the reviewer’s comment.
Point 56 Line: 233 “increasing level of” should be increasing amount of”
Response 56: It is corrected as per the reviewer’s comment.
Point 57 Line: 233 “the level of” should be “concentration of”
Response 57: It is corrected as per the reviewer’s comment.
Point 58 Line: 234 “was decreasing “ should be “decreased”
Response 58: It is corrected as per the reviewer’s comment.
Point 59 Line: 234 “with decreasing level of LF-DDGS” should be “with increasing amount of LF-DDGS”
Response 59: It is corrected as per the reviewer’s comment.
Point 60 Line: 239 “The dressing” should be “Dressing”
Response 60: It is corrected as per the reviewer’s comment.
Point 61 Line: 240 “to be not affected” should be “to not be affected”
Response 61: It is corrected as per the reviewer’s comment.
Point 62 Line: 252 “were higher than the values” should be “were greater than values”
Response 62: It is corrected as per the reviewer’s comment.
Point 63 Line: 253 “the NDF” should be “NDF”
Response 63: It is corrected as per the reviewer’s comment.
Point 64 Line: 253 “are lower than” should be “are less than”
Response 64: It is corrected as per the reviewer’s comment.
Point 65 Line: 255 “the DM” should be “DM”
Response 65: It is corrected as per the reviewer’s comment.
Point 66 Line: 255 “but the TDN” should be “but TDN”
Response 66: It is revised as per the reviewer’s comment.
Point 67 Line: 256 “values were higher” should be “values were greater”
Response 67: It is corrected as per the reviewer’s comment.
Point 68 Line: 260 “the LF-DDGS” should be “LF-DDGS”
Response 68: It is corrected as per the reviewer’s comment.
Point 69 Line: 260 “the LF-DDGS” should be “LF-DDGS”
Response 69: It is corrected as per the reviewer’s comment.
Point 70 Line: 262 “the higher CP” should be “the greater CP”
Response 70: It is revised as per the reviewer’s comment.
Point 71 Line: 263 “the nutrient” should be “nutrient”
Response 71: It is corrected as per the reviewer’s comment.
Point 72 Line: 265 “The increase in ADF” should be “Increased ADF”
Response 72: It is corrected as per the reviewer’s comment.
Point 73 Line: 267 “The Non-Fiber” should be “Non-Fiber”
Response 73: It is corrected as per the reviewer’s comment.
Point 74 Line: 269 “of lower NFC” should be “of decreased NFC”
Response 74: It is corrected as per the reviewer’s comment.
Point 75 Line: 270 “level of ADF” should be “ “amount of ADF”
Response 75: It is revised as per the reviewer’s comment.
Point 76 Line: 271 “and the higher amount of fat” should be “and the greater fat”
Response 76: It is corrected as per the reviewer’s comment.
Point 77 Line: 272 “inhibit the rumen” should be “inhibit rumen”
Response 77: It is corrected as per the reviewer’s comment.
Point 78 Line: 273 “the higher fat” should be “the increased fat”
Response 78: It is revised as per the reviewer’s comment.
Point 79 Line: 273 remove “on it” at the end of the sentence
Response 79: It is corrected as per the reviewer’s comment.
Point 80 Line: 274 “the fat content” should be “fat content”
Response 80: It is corrected as per the reviewer’s comment.
Point 81 Line: 276 “The other” should be “Other”
Response 81: It is corrected as per the reviewer’s comment.
Point 82 Line: 277 “with the increase in the percentage” should be “with increase percentage”
Response 82: It is corrected as per the reviewer’s comment.
Point 83 Line: 279 “the grain mixes” should be “grain mixes”
Response 83: It is corrected as per the reviewer’s comment.
Point 84 Line: 281 “the higher level of” should be “greater concentration of”
Response 84: It is corrected as per the reviewer’s comment.
Point 85 Line: 282 “higher phosphorus” should be “increased phosphorus”
Response 85: It is revised as per the reviewer’s comment.
Point 86 Line: 282 “can lower the cost” should be “decrease cost”
Response 86: It is corrected as per the reviewer’s comment.
Point 87 Line: 284 “level in the goat” should be concentration in the goat”
Response 87: It is corrected as per the reviewer’s comment.
Point 88 Line: 285 “the sulfur” should be “sulfur”
Response 88: It is corrected as per the reviewer’s comment.
Point 89 Line: 289 “the sulfur” should be “sulfur”
Response 89: It is corrected as per the reviewer’s comment.
Point 90 Line: 289 “the grain” should be “grain”
Response 90: It is corrected as per the reviewer’s comment.
Point 91 Line: 292 “do to high NDF” should be “due to more NDF”
Response 91: It is corrected as per the reviewer’s comment.
Point 92 Line: 293 “the LF-DDGS” should be “LF-DDGS”
Response 92: It is corrected as per the reviewer’s comment.
Point 93 Line: 295 “the comparisons” should be “comparisons”
Response 93: It is corrected as per the reviewer’s comment.
Point 94 Line: 296 “the traditional” should be “traditional”
Response 94: It is corrected as per the reviewer’s comment.
Point 95 Line: 299 remove “significant”
Response 95: It is corrected as per the reviewer’s comment.
Point 96 Line: 300 “in DMI in the mid-lactating” should be “for DMI in mid-lactating”
Response 96: It is corrected as per the reviewer’s comment.
Point 97 Line: 302 “difference in the DMI when the reduced fat” should be “difference for DMI when reduced fat”
Response 97: It is corrected as per the reviewer’s comment.
Point 98 Line: 303 “a higher DMI (p < 0.01) in the lactating” should be “an increased DMI (p < 0.01) for lactating”
Response 98: It is revised as per the reviewer’s comment.
Point 99 Line: 306 “a significant increase in DMI” should be “an increase of DMI”
Response 99: It is corrected as per the reviewer’s comment.
Point 100 Line: 307 "the lactating” should be “lactating”
Response 100: It is corrected as per the reviewer’s comment.
Point 101 Line: 308 “the DMI” should be “DMI”
Response 101: It is corrected as per the reviewer’s comment.
Point 102 Line: 310 “no statistical difference” should be “no difference”
Response 102: It is corrected as per the reviewer’s comment.
Point 103 Line: 311 “found in the DMI” should be “found for DMI”
Response 103: It is corrected as per the reviewer’s comment.
Point 104 Line: 311 “as compared” should be “compared”
Response 104: It is corrected as per the reviewer’s comment.
Point 105 Line: 313 “the finishing ration of Boer cross goat” should be “finishing rations of Boer cross goats”
Response 105: It is corrected as per the reviewer’s comment.
Point 106 Line: 318 “the increase in DDGS” should be “increased DDGS”
Response 106: It is corrected as per the reviewer’s comment.
Point 107 Line: 319 remove “significant”
Response 107: It is corrected as per the reviewer’s comment.
Point 108 Line: 326 “increasing level of” should be increasing amount of”
Response 108: It is corrected as per the reviewer’s comment.
Point 109 Line: 328 “increasing level of” should be increasing amount of”
Response 109: It is corrected as per the reviewer’s comment.
Point 110 Line: 330 “Increasing concentration” should be “increased concentration”
Response 110: It is corrected as per the reviewer’s comment.
Point 111 Line: 330 “in the diet of” should be “in diets of”
Response 111: It is corrected as per the reviewer’s comment.
Point 112 Line: 331 “was higher than” should be “was greater than”
Response 112: It is corrected as per the reviewer’s comment.
Point 113 Line: 333 “Any of the feed” should be “Feed”
Response 113: It is corrected as per the reviewer’s comment.
Point 114 Line: 337 “increase in” should be “increased”
Response 114: It is corrected as per the reviewer’s comment.
Point 115 Line: 338 “level of” should be “concentration of”
Response 115: It is corrected as per the reviewer’s comment.
Point 116 Line: 340 “cholesterol level” should be “cholesterol concentration”
Response 116: It is corrected as per the reviewer’s comment.
Point 117 Line: 342 “cholesterol level” should be “cholesterol concentration”
Response 117: It is corrected as per the reviewer’s comment.
Point 118 Line: 344 “of lower fat” should be “of less fat”
Response 118: It is corrected as per the reviewer’s comment.
Point 119 Line: 344 “than higher fat” should be “than more fat”
Response 119: It is corrected as per the reviewer’s comment.
Point 120 Line: 348 “protein level” should be “protein concentration”
Response 120: It is corrected as per the reviewer’s comment.
Point 121 Line: 349 “level of” should be “amount of”
Response 121: It is corrected as per the reviewer’s comment.
Point 122 Line: 350 remove “significant”
Response 122: It is corrected as per the reviewer’s comment.
Point 123 Line: 352 “the serum” should be “serum”
Response 123: It is corrected as per the reviewer’s comment.
Point 124 Line: 352 “in ruminants” should be “of ruminants
Response 124: It is corrected as per the reviewer’s comment.
Point 125 Line: 353 “glucose level” should be “glucose concentration”
Response 125: It is corrected as per the reviewer’s comment.
Point 126 Line: 358 remove “significant”
Response 126: It is corrected as per the reviewer’s comment.
Point 127 Line: 363 remove “significant”
Response 127: It is corrected as per the reviewer’s comment.
Point 128 Line: 371 “DDGS levels” should be “DDGS amount”
Response 128: It is corrected as per the reviewer’s comment.
Point 129 Line: 372 “were higher” should be “were greater”
Response 129: It is corrected as per the reviewer’s comment.
Point 130 Line: 375 “the back fat” should be “back fat”
Response 130: It is corrected as per the reviewer’s comment.
Point 131 Line: 376 “fed with 0%” should be fed 0%”
Response 131: It is corrected as per the reviewer’s comment.
Point 132 Line: 376 “fed with 33” should be fed 33%”
Response 132: It is corrected as per the reviewer’s comment.
Point 133 Line: 376 “than the back” should be “than back”
Response 133: It is corrected as per the reviewer’s comment.
Point 134 Line: 383 “the production” should be “production”
Response 134: It is corrected as per the reviewer’s comment.
Point 135 Line: 383 “the male” should be “male”
Response 135: It is corrected as per the reviewer’s comment.
Point 136 Line: 384 “the prices” should be “prices”
Response 136: It is corrected as per the reviewer’s comment.

Reviewer 2 Report
As you can see below, I started a review. However, the manuscript is so poorly written (grammatically), I am not compelled to make all the necessary corrections for the authors. Thus, I terminated my review. These will be more than minor revisions.
Line Comment
11 “…which has the good nutritional value for livestock…” Poor grammar. Reword.
What is “good” nutritional value for livestock? Good is a relative term and not appropriate here.
12 “In recent years, DDGS produced from the US bioethanol industry are …”
13 Reword to read:
Author Response
Thank you for your comments. We have done extensive revision of our manuscript. We are grateful for these improvements to the presentation of our work and believe that these changes were necessary. We hope this revised submission will achieve the acceptance for publication.
Point 1:
As you can see below, I started a review. However, the manuscript is so poorly written (grammatically), I am not compelled to make all the necessary corrections for the authors. Thus, I terminated my review. These will be more than minor revisions.
Response 1:
The manuscript has been revised extensively and corrected (grammatically).
Point 2: Line 11 “…which has the good nutritional value for livestock…” Poor grammar. Reword. What is “good” nutritional value for livestock? Good is a relative term and not appropriate here.
Response 2: It is revised as per the reviewer’s comment.
Point 3: Line 12 “In recent years, DDGS produced from the US bioethanol industry are …”
Response 3: It is corrected as per the reviewer’s comment.
Reviewer 3 Report
This manuscript explores the possibility of feeding a novel feed ingredient to meat goats. This is a relevant study, although there a some concerns about things in this study that are not adressed.
In the nutrient requirements found in almost all species, it is stated that there should not be more P than Ca in diets. In this study, there is almost twice the amount of P than Ca for a Ca/P of 1/2. THis may perturbate the metabolism of these mineral and affect mainly bone mineralisation. In this study no bone analysis was done. The major concern of Ca/P metabolism, is not discussed deeply in this manuscript. The authors need to adress adequatly the Ca/P of their diets.
L192-196: rephrase to precise that there is not a statistical difference for ADG. There is not even a tendency. Therefore, to write that there is a numerical difference, suggests that there is a difference while in fact there is none, statistically speaking. To say that there is a numerical difference in a scientific paper to lead the reader in a wrong direction is inappropriate. Further, the authors persist when they write in L314-, that others studies "also" found a devrease in ADG. There is no statistical difference in ADG in this study and the authors in the discussion support that there is one. This is a major concerne in science and this manuscript is misleading the readers. Therefore, the result and discussion sections of this manuscript need to be readressed in order for the manuscript to be acceptable for publication.
I have more a less a similar comments for the text relating to figure 1 not showing any significant difference while the authors do not clearly write that. This also needs to be adressed.
Finally corn distillers may concentrate mycotoxins if any are present in the corn grain when harvested. THis concern also needs to be adressed in the MM, result an discussion sections of this manuscript. Major mycotoxins analysis need to be performed on the by-products used in this study, to make sure that other factors did not affect the results.
Author Response
Thank you for your valuable comments on the submitted manuscript. We have completed our revision and revised entire submission based on your comments. We are grateful for these improvements to the presentation of our work and believe that these changes were necessary. We hope that the clarification below and within the text of submission will be able to make this submission acceptable for the publication.
Point 1:
This manuscript explores the possibility of feeding a novel feed ingredient to meat goats. This is a relevant study, although there are some concerns about things in this study that are not addressed.
Response 1:
We tried to address the concerns as per the reviewer’s comments.
Point 2:
In the nutrient requirements found in almost all species, it is stated that there should not be more P than Ca in diets. In this study, there is almost twice the amount of P than Ca for a Ca/P of 1/2. This may perturbate the metabolism of these mineral and affect mainly bone mineralisation. In this study no bone analysis was done. The major concern of Ca/P metabolism, is not discussed deeply in this manuscript. The authors need to address adequately the Ca/P of their diets.
Response 2:
Authors agree with the reviewer’s comment that there is almost twice the amount of P than Ca in grain mixes. However, this Ca and P content is only of the grain mixes which is 50 % of total diet provided to goats. The other 50 % of diet comprised of Bermuda grass hay which approximately contain 0.46 % calcium and 0.20 % phosphorous (as per NRC 2007). And we assume that the calcium and phosphorous ratio was balanced to approximately 1:1 while goats were provided with 50 % of grain mixes and 50 % of Bermuda grass hay.
Point 3
L192-196: rephrase to precise that there is not a statistical difference for ADG. There is not even a tendency. Therefore, to write that there is a numerical difference, suggests that there is a difference while in fact there is none, statistically speaking. To say that there is a numerical difference in a scientific paper to lead the reader in a wrong direction is inappropriate. Further, the authors persist when they write in L314-, that others studies "also" found a decrease in ADG. There is no statistical difference in ADG in this study and the authors in the discussion support that there is one. This is a major concerne in science and this manuscript is misleading the readers. Therefore, the result and discussion sections of this manuscript need to be readressed in order for the manuscript to be acceptable for publication.
Response 3:
The result and discussion section has been rephrased and corrected as per the reviewer’s comments.
Point 4:
I have more a less a similar comments for the text relating to figure 1 not showing any significant difference while the authors do not clearly write that. This also needs to be adressed.
Response 4:
The result has been clearly stated along with p-value as per the reviewer’s comments.
Point 5:
Finally corn distillers may concentrate mycotoxins if any are present in the corn grain when harvested. This concern also needs to be addressed in the MM, result an discussion sections of this manuscript. Major mycotoxins analysis needs to be performed on the by-products used in this study, to make sure that other factors did not affect the results.
Response 5:
We did not perform mycotoxin analysis on the low fat DDGS used in this study. We included this limitation in the discussion section and recommended to do mycotoxin analysis in further researches.
Round 2
Reviewer 3 Report
Thank you for having the manuscript revised. Indeed, the readers can appreciate much improvement.
There is however one point still to be improved. On lines 305-307 you wrote: "The phosphorous content was greater than calcium in grain mixes used in this experiment, but this ratio was expected to be balanced by the BGH (which was 50% of total diet in this experiment) as BGH on average contain 0.46% Ca and 0.20% P [32]."
Your manuscript is dealing with Ca/P. You analyzed these elements for the grain mix as this was the right thing to do. However, the mineral contents of BGH will vary according to maturity at harverst, soil mineral content, pH, etc. Taking NRC mineral values may be fine for a study not dealing with Ca and P. However, since this study, is dealing with a very thin ratio of Ca/P in the total ration, close to 1/1, the authors cannot just write that the ratio "was expected to be balance". Actual BGH analysis needs to be performed and added in the result section and discussed appropriately. Normally, the research team keeps feed samples until the paper is published. Therefore, performing the analysis of the BGH should be feasible and needs to be done, presented and discussed for the manuscript to be acceptable for publication.
Author Response
Point 1:
Thank you for having the manuscript revised. Indeed, the readers can appreciate much improvement.
There is however one point still to be improved. On lines 305-307 you wrote: "The phosphorous content was greater than calcium in grain mixes used in this experiment, but this ratio was expected to be balanced by the BGH (which was 50% of total diet in this experiment) as BGH on average contain 0.46% Ca and 0.20% P [32]."
Your manuscript is dealing with Ca/P. You analyzed these elements for the grain mix as this was the right thing to do. However, the mineral contents of BGH will vary according to maturity at harverst, soil mineral content, pH, etc. Taking NRC mineral values may be fine for a study not dealing with Ca and P. However, since this study, is dealing with a very thin ratio of Ca/P in the total ration, close to 1/1, the authors cannot just write that the ratio "was expected to be balance". Actual BGH analysis needs to be performed and added in the result section and discussed appropriately. Normally, the research team keeps feed samples until the paper is published. Therefore, performing the analysis of the BGH should be feasible and needs to be done, presented and discussed for the manuscript to be acceptable for publication.
Response 1:
Thank you for your comments and suggestions on the submitted manuscript. As suggested, we did the nutrient analysis of Bermuda grass hay and presented values in table 2. We found 0.46 % Ca and 0.19 % P in Bermuda grass hay sample. There were no adverse symptoms observed in experimental goats during 84 days trial period, but when Ca and P content of grain mixes (50%) and BGH (50%) were added and average was calculated for grain mixes, there were still higher P than Ca in total diets with 20 % and 30 % LF-DDGS. So, we recommended to add appropriate amount of Ca supplement to balance the P content in grain mixes. We hope that this revision and clarifications will be able to make this manuscript acceptable for the publication.